# Dietary acrylamide and physical performance tests: A cross-sectional analysis

**Nicola Veronese**[1]*, **Ligia J. Dominguez**[1], **Saverio Ragusa**[1], **Luisa Solimando**[1],
**Lee Smith**[2], **Francesco Bolzetta**[3], **Stefania Maggi**[4‡], **Mario Barbagallo**[1‡]

**1** Geriatric Unit, Department of Internal Medicine and Geriatrics, University of Palermo, Palermo, Italy, **2** The Cambridge Centre for Sport and Exercise Sciences, Anglia Ruskin University, Cambridge, United Kingdom, **3** Azienda Unità Locale Socio Sanitaria 3 "Serenissima", Department of Internal Medicine, Geriatrics Section, Venice, Italy, **4** Aging Branch, Neuroscience Institute, National Research Council, Padua, Italy

‡ These two authors share the last position.
* nicola.veronese@unipa.it

## Abstract

### Background

Dietary acrylamide is found in certain foods, such as deep frying, baking and roasting, and is associated with higher inflammatory and oxidative stress parameters. The association between dietary acrylamide and physical performance has not yet been explored. The aim of the study was to investigate the relationship between dietary acrylamide intake and physical performance tests in a large cohort of North American individuals affected by knee osteoarthritis or at high risk for this condition.

### Methods

Dietary acrylamide intake was obtained through a food frequency questionnaire and reported in quartiles and as an increase in deciles. Physical performance was explored using the 20-meter usual pace test, the 400-meter walking distance, and the chair stands time. The association between dietary acrylamide and physical performance tests was explored using linear regression analysis, adjusted for potential confounders.

### Results

4,436 participants (2,578 women, mean age: 61.3) were enrolled. People in the highest quartile of dietary acrylamide reported significantly longer 20-meter walking (15.53±3.32 vs. 15.15±2.91 s), 400-meter walking (312±54 vs. 305±58 s) and chair stands (11.36±4.08 vs. 10.67±3.50 s) times than their counterparts in Q1. In adjusted linear regression analyses, each increase in one decile in dietary acrylamide was associated with a longer time in walking for 20 meters (beta = 0.032; 95%CI: 0.016–0.048; p = 0.04), 400 meters (beta = 0.048; 95%CI: 0.033–0.063; p = 0.002) and chair stands (beta = 0.016; 95%CI: 0.005–0.037; p = 0.04) times.

**Data Availability Statement:** Data from the Osteoarthritis Initiative (OAI) database (https://www.niams.nih.gov/grants-funding/funded-research/osteoarthritis-initiative) were used for this

research. These data can be freely downloaded without any special privilege.

**Funding:** The authors received no specific funding for this work.

**Competing interests:** The authors have declared that no competing interests exist.

## Conclusion

Higher dietary acrylamide intake was significantly associated with poor physical performance, also after accounting for potential confounders, suggesting a role for this food contaminant as a possible risk factor for sarcopenia.

## Introduction

Acrylamide is a vinyl monomer principally derived from chemical industries to produce polymers for water treatment, oil drilling, paper making and mineral processing [1]. Acrylamide was firstly evaluated by the International Agency for Research on Cancer and identified as potentially carcinogenic to humans [2]. As a food component, acrylamide can be formed during thermal processing of carbohydrate-rich foods, especially deep-frying, oven-baking, and roasting [2, 3].

The amount of acrylamide in cooked foods is determined by cooking temperature and time and the quantity of reducing sugar and asparagine in raw foods [4]. The World Health Organization (WHO) attempted to evaluate the average exposure to acrylamide in foods, however this is difficult to measure with traditional instruments available in nutritional research [5]. Acrylamide is included by the International Agency for Research on Cancer into the group 2A, i.e., a probable human carcinogenic [6]. From a metabolic point of view, acrylamide can be metabolized using two major pathways. The first one is the pathway involving cytochrome P450 2E1 (CYP2E1)-mediated phase: through this way the oxidative metabolite glycidamide is consequently formed. Glycidamide can react with DNA to create DNA adducts. The second pathway of acrylamide metabolism is the direct conjugation with reduced glutathione (GSH). Acrylamide and glycidamide are also able to be associated with albumin or other plasma proteins. Both acrylamide and glycidamide can react with the N-terminal valine residues of Hemoglobin to form Hb adducts of acrylamide, that are widely used as biomarkers for acrylamide exposure [7].

Recent literature has suggested that acrylamide exposure might lead to an increased risk of oxidative stress as shown by a significant increase in reactive oxygen species (ROS) and malondialdehyde (MDA) levels and glutathione (GSH) reduction. Inflammatory response was observed based on dose-dependent levels of pro-inflammatory cytokines tumor necrosis factor-$\alpha$ (TNF-$\alpha$) and interleukin 6 (IL-6) [8]. In addition, acrylamide activated nuclear transcription factor E2-related factor 2 (Nrf2) and nuclear factor-$\kappa$B (NF-$\kappa$B) signaling pathways were also observed [8]. Moreover, analogs effects on inflammation parameters were reported in another study [9].

At the same time, there is increasing attention to inflammation and oxidative stress as potential risk factors for poor physical performance and its consequences. It is reported that people with sarcopenia, for example, had significantly higher serum levels of both inflammatory [10] and oxidative stress markers [11]. Similarly, frailty, a typical condition of older persons characterized by muscle loss and reduced reserve to stressor events, is associated with higher inflammatory and oxidative stress markers [12, 13]. In this regard, diet seems to play an important role since an unhealthy diet has been observed to significantly increase inflammatory markers in human beings [14]. Unfortunately, studies reporting the potential association between dietary acrylamide and physical performance tests are not available.

Given this background, the aim of the present study was to investigate the relationship between dietary acrylamide intake and physical performance tests in a large cohort of North American individuals affected by knee osteoarthritis or at high risk for this condition.

## Materials and methods

### Data source and subjects

Data from the Osteoarthritis Initiative (OAI) database (https://www.niams.nih.gov/grants-funding/funded-research/osteoarthritis-initiative) were used for this research. These data can be freely downloaded without any special privilege. Participants were included from four different sites in the United States of America (Baltimore, MD; Pittsburgh, PA; Pawtucket, RI; and Columbus, OH) between February 2004 and May 2006. All participants provided written informed consent. In the OAI study, we identified people who either: (1) had knee OA with knee pain for a 30-day period in the past 12 months or (2) were at high risk of developing knee OA [15] with data collected during baseline and screening evaluations. Therefore, the OAI is representative of only people affected by knee OA and not general population. The OAI study was given full ethical approval by the institutional review board of the OAI Coordinating Center, at University of California in San Francisco.

### Exposure

Dietary acrylamide intake was obtained through a food frequency questionnaire (FFQ) recorded during the baseline visit of the OAI. The product yearly frequency by portions' size, contained in the FFQ, was categorized into standard portions (median value) [16] and then grams. The presence of acrylamide in foods was estimated using 2015 data from the Food and Drugs Administration (FDA) website (https://www.fda.gov/food/chemicals/survey-data-acrylamide-food). The general limit of quantitation (LOQ) of the method used by the FDA is 10 mcg. The data for this work regarding dietary acrylamide were reported as mcg consumed in one year by each subject. We divided the participants into gender-specific quartiles for descriptive purposes and in deciles for the analyses regarding linear regression.

### Outcomes

In the OAI study, physical performance was assessed using several standardized tests, commonly used in geriatric medicine: (i) 20-meter at a usual pace that was repeated two times, using the best time for the aims of the analyses [17]; (ii) 400-meter walking distance was performed according to an evaluated protocol to assess mobility disability and it consists of 10 laps over a 20 m course marked by two cones [18]; (iii) chairs stands, repeated five times for each trial (n = 2) using the best time for the analyses [19]. For all these parameters, reported in seconds, higher values indicate worse performance.

### Covariates

Several covariates were identified as potential confounding factors. These included: age; gender; total calorie intake (in Kcal); body mass index (BMI); race; smoking habit; educational attainment level (college or higher vs. others); yearly income ($<$ or $\geq$ \$50,000 or missing data); the modified Charlson Comorbidity Index score [20]; physical activity level, measured with the PASE (physical activity scale for the elderly) [21]; presence of knee osteoarthritis (OA), defined as the combination in the clinical reporting and assessment of pain and stiffness (i.e. pain, aching or stiffness in or around the knee on most days during the last year), and radiographical OA on the baseline fixed flexion radiograph based on the presence of tibiofemoral osteophytes (correspondent to Osteoarthritis Research Society International atlas grades 1–3, clinical center reading) [22].

## Statistical analyses

Continuous variables were normally distributed according to the Kolmogorov-Smirnov test. Therefore, data were shown as means and standard deviation values (SD) for quantitative measures. Percentages were used for discrete variables. Levene's test was used to test the homoscedasticity of variances and, if its assumption was violated, Welch's ANOVA was used. P-values were calculated using the Jonckheere-Terpstra test [23] for continuous variables and the Mantel-Haenszel Chi-square test for categorical variables.

For assessing the relationship between dietary acrylamide intake and physical performance tests, we used linear regression analysis. The basic adjusted model included only age and gender, whilst the fully adjusted model included all the aforementioned covariates. Multi-collinearity among covariates was assessed using variance inflation factor (VIF) [24], taking a cut-off of 2 as the criterion for exclusion. Standardized betas and 95% confidence intervals (CI) were reported to estimate the strength of the associations between dietary acrylamide intake and physical performance tests.

To test the robustness of our results, we stratified our results considering the presence (or not) of knee OA since knee OA is highly present in this population and may increase the risk of sarcopenia, but the p-values for the interaction between dietary acrylamide, in deciles, and the presence of knee OA were >0.05.

All analyses were performed using the SPSS 20.0 for Windows (SPSS Inc., Chicago, Illinois). All statistical tests were two-tailed and statistical significance was assumed for a p-value <0.05.

## Results

### Study participants

In 4,796 initially included individuals, 243 reported < than 500 or > 5,000 Kcal (not reliable calorie intake) or did not have any information regarding FFQ and 117 had not sufficient data regarding physical performance tests.

### Descriptive analyses

Altogether, 4,436 participants (of them 2,578 women) with a mean age of 61.3 ±9.1 (range: 45–79) years were included in the analysis. The yearly mean acrylamide intake was estimated in 13,985±23,863 (range: 0–364,438) mcg.

The baseline characteristics, by dietary acrylamide intake divided into quartiles, are shown in **Table 1**. People having a greater dietary acrylamide intake (Q4) were significantly younger, less educated, had a significantly higher calorie intake and were more frequently obese, but more physically active than the participants introducing less acrylamide with their diet (Q4) (**Table 1**). Regarding health issues, participants with the highest amount of acrylamide did not differ in term of Charlson comorbidity index (p = 0.76), but they reported a significantly higher prevalence of diabetes and lower prevalence of cancer (p = 0.003) than their counterparts (**Table 1**).

People in the highest quartile of dietary acrylamide reported significantly longer 20-meter walking (15.53±3.32 vs. 15.15±2.91 s, p<0.0001), 400-meter walking (312±54 vs. 305±58 s, p = 0.005) and chair stands (11.36±4.08 vs. 10.67±3.50 s, p<0.0001) times than their counterparts in Q1 (**Table 1**).

### Linear regression analysis

**Table 2** shows the linear regression analysis, taking dietary acrylamide as increases in deciles as exposure variable and physical performances tests as outcomes. Increasing levels of dietary

**Table 1. Descriptive characteristics by dietary acrylamide intake[1].**

| Parameter | Q1 (n = 1127) | Q2 (n = 1085) | Q3 (n = 1140) | Q4 (n = 1084) | p-value |
|---|---|---|---|---|---|
| Age (SD) | 63.8 (9.2) | 61.9 (9.0) | 60.9 (9.0) | 58.5 (8.7) | <0.0001 |
| Female gender (%) | 57.5 | 59.0 | 58.9 | 57.1 | 0.86 |
| Calorie intake (Kcal) | 1213 (460) | 1299 (498) | 1415 (500) | 1740 (662) | <0.0001 |
| PASE (SD) | 156 (81) | 161 (80) | 160 (84) | 166 (83) | 0.04 |
| BMI (Kg/m$^2$) | 27.3 (4.6) | 28.4 (4.5) | 29.0 (4.8) | 30.1 (4.9) | <0.0001 |
| Presence of knee OA (radiological diagnosis) (%) | 25.1 | 28.8 | 29.6 | 32.6 | <0.0001 |
| Charlson comorbidity index (points) | 0.40 (0.89) | 0.37 (0.84) | 0.38 (0.82) | 0.40 (0.82) | 0.76 |
| Cardiovascular disease (%) | 4.0 | 3.5 | 4.5 | 4.0 | 0.65 |
| Asthma (%) | 9.3 | 8.7 | 8.1 | 9.0 | 0.65 |
| Chronic obstructive pulmonary disease (%) | 2.1 | 2.2 | 2.0 | 2.6 | 0.56 |
| Ulcer disease (%) | 2.7 | 2.1 | 3.6 | 2.4 | 0.80 |
| Diabetes (%) | 6.4 | 6.6 | 8.1 | 9.4 | 0.003 |
| Poor renal function (%) | 1.1 | 1.5 | 1.2 | 1.7 | 0.33 |
| Cancer (%) | 5.4 | 3.3 | 3.5 | 2.7 | 0.003 |
| Yearly income (>50,000 $) | 59.2 | 61.0 | 61.1 | 54.7 | 0.006 |
| Whites (%) | 87.0 | 82.9 | 81.0 | 70.1 | <0.0001 |
| College or higher (%) | 35.8 | 32.3 | 27.5 | 25.8 | <0.0001 |
| Current/previous smokers (%) | 53.0 | 53.5 | 52.7 | 50.8 | 0.29 |
| 20-meter walking time | 15.15 (2.91) | 14.95 (2.46) | 15.33 (2.97) | 15.53 (3.32) | <0.0001 |
| 400-meter total time | 305 (58) | 303 (54) | 308 (60) | 312 (54) | 0.005 |
| Chair stands time | 10.67 (3.50) | 10.84 (3.34) | 10.92 (3.85) | 11.36 (4.08) | <0.0001 |

[1] **Abbreviations:** BMI: body mass index; OA: osteoarthritis; PASE: physical activity scale for elderly SD: standard deviation.

acrylamide were associated with worse performance in physical performance tests after adjusting for age and gender. After including the effect of eleven potential confounders, each increase in one decile in dietary acrylamide was associated with a longer time in walking for 20 meters (beta = 0.032; 95%CI: 0.016–0.048; p = 0.04), 400 meters (beta = 0.048; 95%CI: 0.033–0.063; p = 0.002) and chair stands (beta = 0.016; 95%CI: 0.005–0.037; p = 0.04) times (**Table 2**).

## Discussion

In the present study that includes a large cohort of US adults affected by knee OA or at high risk for this condition, we found that people with higher dietary acrylamide intake have worse physical performance tests than people introducing less amounts of acrylamide. These findings

**Table 2. Association between acrylamide intake and physical performance tests.**

| Physical performance test | Basic-adjusted estimates | | | Fully adjusted estimates | | |
|---|---|---|---|---|---|---|
| | beta | 95% CI | p-value | beta | 95% CI | p-value |
| 20-meter walking time | 0.104 | 0.088–0.120 | <0.0001 | 0.032 | 0.016–0.048 | 0.04 |
| 400-meter total time | 0.113 | 0.097–0.129 | <0.0001 | 0.048 | 0.033–0.063 | 0.002 |
| Chair stands time | 0.128 | 0.107–0.149 | <0.0001 | 0.016 | 0.005–0.037 | 0.04 |

[1] Data are reported as standardized betas with their 95% confidence intervals and correspondent p-values. Basic-adjusted model includes age and gender; fully adjusted model includes, other than age and sex: smoking status, presence and of comorbidities, educational level, ethnicity, body mass index, yearly income, total energy intake, physical activity scale for elderly values, presence of radiological knee osteoarthritis.

remained unaltered, after adjustment for several potential confounders, indicating a possible role of dietary acrylamide in poor physical performance.

Of importance, participants having a higher dietary acrylamide intake were significantly younger, less educated, had a significantly higher calorie intake and were more frequently obese than those introducing less acrylamide with their diet. These findings were expected, since in this study a high acrylamide intake reflects a propensity to an unhealthy diet, rich in fried and roasted food, common in North American obese people and in younger populations [25]. It is important to note that people consuming more acrylamide with their diet are, in mean, 5 years younger than those with lower intakes, but paradoxically report worse physical performance tests indicating an important role of unhealthy diet in predicting poor physical performance.

To the best of our knowledge, this is the first study evaluating the relationship between the dietary intake of acrylamide and physical performance, indicating the need for future research in this field. However, we can speculate regarding these data, at least from a pathophysiological point of view. First, it is possible that reactive oxygen species may partially explain the relationship between dietary acrylamide intake and poor physical performance. Indeed, a previous study observed that acrylamide exposure was associated with an increase in reactive oxygen species [8]. Second, in experimental animal models, acrylamide increases inflammatory and pro-apoptosis markers [26], further contributing to muscle loss and, therefore, to poor physical performance.

The findings from this study should be interpreted considering its limitations. First, the OAI included only participants with knee osteoarthritis or at high risk of this condition, likely introducing a selection bias. Second, the cross-sectional nature of the study can introduce a potential reverse causation issue, i.e., people with worse physical performance tests can report higher dietary acrylamide intake (e.g., they are more obese). Finally, in the OAI, no muscle mass quality assessment was carried out and these data could be important for better understanding the association between acrylamide and poor physical performance.

In conclusion, higher dietary acrylamide intake was significantly associated with poor physical performance, also after accounting for potential confounders, suggesting a role for this food contaminant as a possible risk factor for sarcopenia. Future longitudinal studies are however needed to confirm or refute the present findings.

## Author Contributions

**Conceptualization:** Francesco Bolzetta.

**Data curation:** Lee Smith, Mario Barbagallo.

**Formal analysis:** Nicola Veronese, Lee Smith.

**Supervision:** Stefania Maggi.

**Writing – original draft:** Nicola Veronese, Saverio Ragusa, Luisa Solimando, Francesco Bolzetta.

**Writing – review & editing:** Ligia J. Dominguez, Stefania Maggi, Mario Barbagallo.

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
