## [Decision Letter · Decision Letter 0]

20 Sep 2021

PONE-D-21-24104DIETARY ACRYLAMIDE AND PHYSICAL PERFORMANCE TESTS:  A CROSS-SECTIONAL ANALYSISPLOS ONE

Dear Dr. Veronese,

Thank you for submitting your manuscript to PLOS ONE. After careful consideration, we feel that it has merit but does not fully meet PLOS ONE’s publication criteria as it currently stands. Therefore, we invite you to submit a revised version of the manuscript that addresses the points raised during the review process.

I am very much thankful to the reviewers for their deep and thorough review.Some minor revisions are required. Please check the reviewers’ comments.It remains unclear why there is a significant delay in publishing their data. ==============================

We look forward to receiving your revised manuscript.

Kind regards,

Sıdıka Bulduk, Prof. Dr.

Academic Editor

PLOS ONE

Journal Requirements:

2. In your Methods section, please provide additional information about the participant recruitment method and the demographic details of your participants. Please ensure you have provided sufficient details to replicate the analyses such as:

a) a description of any inclusion/exclusion criteria that were applied to participant selection,

b) a statement as to whether your sample can be considered representative of a larger population

“The OAI is a public-private partnership comprised of five contracts (N01-AR-2-2258; N01-AR-2-2259; N01-AR-2-2260; N01-AR-2-2261; N01-AR-2-2262) funded by the National Institutes of Health, a branch of the Department of Health and Human Services, and conducted by the OAI Study Investigators. Private funding partners include Merck Research Laboratories; Novartis Pharmaceuticals Corporation, GlaxoSmithKline; and Pfizer, Inc. Private sector funding for the OAI is managed by the Foundation for the National Institutes of Health. This manuscript was prepared using an OAI public use data set and does not necessarily reflect the opinions or views of the OAI investigators, the NIH, or the private funding partners.”

Reviewers' comments:

Reviewer's Responses to Questions

**Comments to the Author**

1. Is the manuscript technically sound, and do the data support the conclusions?

Reviewer #1: Yes

Reviewer #2: Partly

2. Has the statistical analysis been performed appropriately and rigorously? 

Reviewer #1: Yes

Reviewer #2: Yes

3. Have the authors made all data underlying the findings in their manuscript fully available?

Reviewer #1: Yes

Reviewer #2: No

4. Is the manuscript presented in an intelligible fashion and written in standard English?

Reviewer #1: Yes

Reviewer #2: Yes

5. Review Comments to the Author

Reviewer #1: I reviewed the "Dietary Acrylamide and Physical Performance Tests: A Cross-Sectional Analysis" entitled manuscript and my suggestions are listed below.

General comments: Acrylamide is an important contaminant, and the amount of acrylamide that people take in diet is very important for public health. The topic of the paper is interesting.

References should be written according to the journal rules. Punctuation marks should be corrected in the writing of references in the text. At the end of the sentence, the reference should be shown in parentheses, followed by a period. Abbreviations and explanations should be given as footnotes in the tables.

In which group is acrylamide classified by the International Agency for Research on Cancer? This information and metabolism of acrylamide should be given in the introduction. There are typos in the paper. Some corrections are listed below.

ABSTRACT

Background:

• “Dietary acrylamide is present is several deep-frying, oven-baking, and roasting foods and associated with higher inflammatory and oxidative stress parameters.” Should be written as “Dietary acrylamide is found in certain foods, such as deep frying, baking and roasting, and is associated with higher inflammatory and oxidative stress parameters.”

Methods:

• “as increase in” should be written as “as an increase in”

• “using a linear regression” should be written as “using linear regression”

INTRODUCTION

• “Moreover, analogues” should be written as “Moreover, analogs”

• “glutathione (GSH) reduction.(6)” and “interleukin 6 (IL-6). (6)” reference (6) should be removed. In the continuation of these two sentences in the same paragraph “6th” reference has already been given.

METHODS

Exposure:

• “We divided the participants in gender-specific” should be written as “We divided the participants into gender-specific”

Outcomes:

• “20-meter at usual pace” should be written as “20-meter at a usual pace”

Covariates:

• “presence of knee OA” acronym should be written as full terminology in the first instance.

Statistical analysis:

• “we used a linear regression analysis.” Should be written as “we used linear regression analysis.”

RESULTS

Descriptive analyses:

• “by dietary acrylamide intake divided in”, “in” should be written as “into”

• “had a significant higher calorie intake”, “significant” should be written as “significantly”

• “but more physical active”, “physical” should be written as “physically”

Table 1:

“Charlson comoribidity index (points)”, “comoribidity” should be corrected as “comorbidity”

DISCUSSION

• “had a significant higher calorie intake”, “significant” should be written as “significantly”

• “indicating the need of future research”, “of” should be written as “for”

Reviewer #2: The link for the data is not working.

Only Dietary acrylamide intake was obtained through a food frequency questionnaire and correlated with the Physical performance. No other details regarding to the health parameters were shared or discussed. It would strengthen the results.

6. PLOS authors have the option to publish the peer review history of their article (what does this mean?). If published, this will include your full peer review and any attached files.

Reviewer #1: No

Reviewer #2: No

---

## [Author Response · Author response to Decision Letter 0]

5 Oct 2021

R: The name of the files are in the correct form. 

2. In your Methods section, please provide additional information about the participant recruitment method and the demographic details of your participants. Please ensure you have provided sufficient details to replicate the analyses such as:

a) a description of any inclusion/exclusion criteria that were applied to participant selection,

b) a statement as to whether your sample can be considered representative of a larger population

R: Added this information. 

R: Added, as suggested. 

“The OAI is a public-private partnership comprised of five contracts (N01-AR-2-2258; N01-AR-2-2259; N01-AR-2-2260; N01-AR-2-2261; N01-AR-2-2262) funded by the National Institutes of Health, a branch of the Department of Health and Human Services, and conducted by the OAI Study Investigators. Private funding partners include Merck Research Laboratories; Novartis Pharmaceuticals Corporation, GlaxoSmithKline; and Pfizer, Inc. Private sector funding for the OAI is managed by the Foundation for the National Institutes of Health. This manuscript was prepared using an OAI public use data set and does not necessarily reflect the opinions or views of the OAI investigators, the NIH, or the private funding partners.”

R: Removed the funding information from the manuscript, leaving only in the cover letter. No funding was received by the Authors for this work. 

R: The data of the OAI are freely accessible at: https://www.niams.nih.gov/grants-funding/funded-research/osteoarthritis-initiative, after making a simple login. Therefore, we will not change what declared before. 

R: Done. 

R: The reference list is correct. 

 

Reviewer #1: I reviewed the "Dietary Acrylamide and Physical Performance Tests: A Cross-Sectional Analysis" entitled manuscript and my suggestions are listed below.

General comments: Acrylamide is an important contaminant, and the amount of acrylamide that people take in diet is very important for public health. The topic of the paper is interesting.

R: We would like to sincerely thank the Reviewer for her/his appreciation for our manuscript. We have tried to further improve it through the comments of Reviewer 1 and 2. 

References should be written according to the journal rules. Punctuation marks should be corrected in the writing of references in the text. At the end of the sentence, the reference should be shown in parentheses, followed by a period. Abbreviations and explanations should be given as footnotes in the tables.

R: We have addressed all these important points, as suggested. 

In which group is acrylamide classified by the International Agency for Research on Cancer? This information and metabolism of acrylamide should be given in the introduction. 

R: We sincerely thank the Reviewer for this comment. We have now added this explanation in the Introduction section, as follows: 

“Acrylamide is included by the International Agency for Research on Cancer into the group 2A, i.e., a probable human carcinogenic [6]. From a metabolic point of view, acrylamide can be metabolized using two major pathways. The first one is the pathway involving cytochrome P450 2E1 (CYP2E1)-mediated phase: through this way the oxidative metabolite glycidamide is consequently formed. Glycidamide can react with DNA to create DNA adducts. The second pathway of acrylamide metabolism is the direct conjugation with reduced glutathione (GSH). Acrylamide and glycidamide are also able to be associated with albumin or other plasma proteins. Both acrylamide and glycidamide can react with the N-terminal valine residues of Hemoglobin to form Hb adducts of acrylamide, that are widely used as biomarkers for acrylamide exposure [7].”

There are typos in the paper. Some corrections are listed below.

ABSTRACT

Background:

• “Dietary acrylamide is present is several deep-frying, oven-baking, and roasting foods and associated with higher inflammatory and oxidative stress parameters.” Should be written as “Dietary acrylamide is found in certain foods, such as deep frying, baking and roasting, and is associated with higher inflammatory and oxidative stress parameters.”

Methods:

• “as increase in” should be written as “as an increase in”

• “using a linear regression” should be written as “using linear regression”

INTRODUCTION

• “Moreover, analogues” should be written as “Moreover, analogs”

• “glutathione (GSH) reduction.(6)” and “interleukin 6 (IL-6). (6)” reference (6) should be removed. In the continuation of these two sentences in the same paragraph “6th” reference has already been given.

METHODS

Exposure:

• “We divided the participants in gender-specific” should be written as “We divided the participants into gender-specific”

Outcomes:

• “20-meter at usual pace” should be written as “20-meter at a usual pace”

Covariates:

• “presence of knee OA” acronym should be written as full terminology in the first instance.

Statistical analysis:

• “we used a linear regression analysis.” Should be written as “we used linear regression analysis.”

RESULTS

Descriptive analyses:

• “by dietary acrylamide intake divided in”, “in” should be written as “into”

• “had a significant higher calorie intake”, “significant” should be written as “significantly”

• “but more physical active”, “physical” should be written as “physically”

Table 1:

“Charlson comoribidity index (points)”, “comoribidity” should be corrected as “comorbidity”

DISCUSSION

• “had a significant higher calorie intake”, “significant” should be written as “significantly”

• “indicating the need of future research”, “of” should be written as “for”

R: Thank you so much for all these comments and for your careful reading. We have now corrected all these typos, as suggested. 

Reviewer #2: The link for the data is not working.

R: Thank you for your careful reading. We have used a previous version of the website. In the Revised version, you can find the correct link. 

Only Dietary acrylamide intake was obtained through a food frequency questionnaire and correlated with the Physical performance. No other details regarding to the health parameters were shared or discussed. It would strengthen the results.

R: We would like to sincerely thank the Reviewer. The results were already adjusted for the Charlson comorbidity index that included relevant health issues and medical conditions. However, we have reported the prevalence of some important medical conditions for better understanding the prevalence of some common diseases that could associated with dietary acrylamide and poor physical performance.

---

## [Editor Report · Decision Letter 1]

18 Oct 2021

DIETARY ACRYLAMIDE AND PHYSICAL PERFORMANCE TESTS:  A CROSS-SECTIONAL ANALYSIS

PONE-D-21-24104R1

Dear Dr. Veronese,

We’re pleased to inform you that your manuscript has been judged scientifically suitable for publication and will be formally accepted for publication once it meets all outstanding technical requirements.

Kind regards,

Sıdıka Bulduk, Prof. Dr.

Academic Editor

PLOS ONE
---

## [Editor Report · Acceptance letter]

22 Oct 2021

PONE-D-21-24104R1 

Dietary acrylamide and physical performance tests: a cross-sectional analysis 

Dear Dr. Veronese:

I'm pleased to inform you that your manuscript has been deemed suitable for publication in PLOS ONE. Congratulations! Your manuscript is now with our production department. 

Kind regards, 

on behalf of

Dr. Sıdıka Bulduk 

Academic Editor

PLOS ONE